# Land- and Water-Based Adaptive Farming Practices to Cope with Waterlogging in Variably Elevated Homesteads

**Md. Moshiur Rahman** [1,2,*], **Tapan Kumar Chakraborty** [3], **Abdullah Al Mamun** [3] **and Victor Kiaya** [4]

1   Fish Conservation and Culture Lab, Department of Biological & Agricultural Engineering, University of California, 17501 Byron Hwy, Byron, Davis, CA 94514, USA
2   Fisheries and Marine Resource Technology Discipline, Khulna University, Khulna 9208, Bangladesh
3   Food Security Livelihood and DRR, Action Against Hunger, Dhaka 1212, Bangladesh
4   Regional Food Security & Livelihoods Programme, Action Against Hunger, 93100 Paris, France
*   Correspondence: momrahman@ucdavis.edu; Tel.: +1-346-399-3832

**Abstract:** Waterlogging is a major problem in the south-western region of Bangladesh; this study was conducted in the eight most affected areas in order to enhance agricultural production by applying Land- and Water-based adaptive and alternative Farming Practices (LWFP). The study was designed to support target (research) farmers by raising one part of their homestead to use for living and agricultural farming, with the other part excavated to store rainwater and use for aquaculture. The study selected two groups of control farmers: those with ponds and those without. The study was conducted in two phases (i.e., phase 1—pilot phase and phase 2—extended phase), with each year divided into three cropping seasons: summer, rainy, and winter. The study found that the research farmers' income was significantly higher from vegetables (both pilot and extended phases: $p < 0.001$), dike crops (both pilot and extended phases: $p < 0.001$), fish (both pilot and extended phases: $p < 0.001$), livestock (pilot phase: $p < 0.01$ and extended phase: $p < 0.001$), and poultry (pilot phase: $p < 0.05$ and extended phase: $p < 0.001$) compared to the control farmers. Moreover, the research supported the empowerment of women, which was not found in the control farms. Overall, the research program was embraced by the local communities as a very successful model. Furthermore, the study showed how waterlogging marginally affects very poor people, and that they can cope with this severe problem by adopting various farming practices. Therefore, the application of this research approach is suggested for similarly affected areas.

**Keywords:** climate change; integrated farming; adaptive agriculture; flood effects; livelihoods

## 1. Introduction

The Intergovernmental Panel on Climate Change (IPCC) has reported that the frequency of heatwaves, number of extreme precipitation events, and occurrences of frequent coastal flooding have increased due to climate change in many parts of the world [1]. The IPCC has already predicted that risks associated with extreme events will continue to increase as the global mean temperature rises. It is expected that the Earth's temperature could rise 1.5 °C [2]. Among these risks, frequent and heavy precipitation is one of the most destructive climate change variables. Data from the most affected countries show that extreme precipitation can cause floods and landslides, which have hit many regions in Africa, South and Southeast Asia, and South America [3]. Extreme precipitation is expected to increase as global warming intensifies the global hydrological cycle. Therefore, single precipitation events are expected to increase at a higher rate than global mean changes in total precipitation [4]. Moreover, Wasko and Sharma [5] have predicted that warmer temperatures due to climate change could increase the magnitude and frequency of floods. This has been supported by the increasing evidence for the link between extreme El Niño events and global warming; the occurrence of such events could double in the future due to climate change [6].

Bangladesh has been identified as one of the most climate change-affected countries in the world [3]; in particular, the south-west coastal region is the most disaster-prone, being highly vulnerable to climate change-related risks [7–9]. Various studies and reports have documented that the south-west coastal districts (e.g., Khulna, Satkhira, Jessore, and Barguna) are the most affected areas of Bangladesh; these areas have been experiencing problems with 'waterlogging' since the early 1980s [7,8,10]. The term 'waterlogging' refers to the inundation of an area every year with long-term flooding (up to six months) after heavy rainfall. The major causes of waterlogging in this region include extreme rainfall, riverbeds rising due to siltation, sea-level rises, unplanned urbanization, and blocking of drainage systems through different infrastructural developments [7,10]. Every year, water-logging engulfs tens of thousands of hectares, with a devastating effect on livelihoods and quality of life. Robson [7] has reported that approximately 68,194 ha within eight upazilas (there are five administrative tiers of local government in Bangladesh: Division, District, Upazila, Union and Ward; the upazilas are the third-lowest tier of regional administration in Bangladesh) in the Khulna, Satkhira, and Jessore districts were waterlogged in 2013. The number was 73,698 ha in 2009, and 50,924 ha in 2006. In another study, Rahman et al. [11], using satellite images, showed that over the years the waterlogged area had increased from 865 ha in 1999 to 19,467 ha in 2008, making it a regular phenomenon for the hundreds of villages adjacent to the Kopadak river in Jessore and Satkhira districts. Hassan and Islam [12] detected the waterlogged area through Landsat imagery from 1972, 1989, and 2014 in Jessore district, where about 32,830 ha (13% of the total land) was identified as a waterlogged area. A severe waterlogging problem was reported by Paul et al. [13] in the Bhabodaoh area of Jessore district from October 2005 to November 2006; this inundated area was recorded as about 18,100 ha in September 2006.

The south-west region of Bangladesh is a part of and has been formed by an active delta system. Therefore, the soil is relatively fertile, allowing for medium to high-level intensified agriculture and multiple crop types, which are cultivated throughout the year during three main seasons. Several studies have shown that the overall direct impact of waterlogging on livelihoods in this region is massive [7,14,15]. A recent study revealed that more than 650,000 people and around 128,000 ha of crop land (out of a total of 200,000 ha) were affected by monsoon flooding and subsequent waterlogging in these areas during 2011 [10]. Rahman et al. [11] reported that about 101,800 people were affected by waterlogging in the Kopadak river basin of Jessore and Satkhira districts during 2003, and that this number increased to 845,000 people in 2008. Much variation can be seen in the number of affected people within the different districts and upazilas among the different reports. However, most previous studies have documented that the highest number of people affected were in the Tala, Kalaroa, and Sadar upazilas of Satkhira district and the Keshobpur upazila of Jessore district [7,10,14,16].

Waterlogging affects people directly or indirectly in many ways: it damages their livelihood assets such as houses, roads, homestead gardens, plants, domestic animals, and birds; it inundates ponds, and thereby damages fish production; it damages and shrinks valuable cultivated croplands; it disrupts transportation; and it destroys many other valuable assets [7,10,16]. Thus, affected people become homeless, jobless, food deficient, malnourished, deprived of education and health facilities, isolated from the community, and insecure in every way. Considering the conditions of these vulnerable communities, much research has been done on the problem [7,14,15,17]. Most of these projects were conducted in order to detect waterlogged areas [7,11,12] and to ascertain the socio-economic conditions of the affected people [7,14,15], the physical infrastructure of affected regions and possible reconstruction [13,15], the probable causes and remediation of waterlogging [7,10], and how to cope with waterlogging for livelihood purposes [7,15]. Unfortunately, very few studies [10,16,17] have been carried out to demonstrate how these vulnerable communities can cope with waterlogging through agricultural multi-crop production using their very limited resources. Because agriculture is the main source of food industry in this area, it has been identified as one of the most critical sectors

for increasing food security and reducing poverty, underdevelopment, and inequality in order to achieve United Nations Sustainable Development Goals (SDGs) 1 and 2 of the 2030 Agenda [18–20]. Therefore, the present research was taken up to help in producing multiple crops throughout the year using only limited homestead resources by applying sustainable, adaptive, and alternative agricultural technologies in order to cope with the severe waterlogging problem.

## 2. Materials and Methods

### 2.1. Study Area

Considering the extent, frequency, and magnitude of the waterlogging problem, the time limitations and budget allocation for the work, and the availability of assistance from partner organizations, this study was carried out in two phases. In phase 1 (hereafter called the 'pilot phase'), eight (8) unions (the forth lowest tier of regional administration in Bangladesh) in the Satkhira (6) and Jessore (2) districts were selected for a baseline survey in 2016; this was then applied for research purposes in 2017 (see Table 1 and Figure 1). In phase 2 (hereafter called the 'extended phase'), just two (2) unions in the Satkhira district were selected for an extended study; the baseline survey for this phase was conducted in 2017, and the research work was carried out from 2018 to 2020 (see Table 1 and Figure 1). These areas were selected based on previous studies which showed that the selected unions had been severely affected by waterlogging within the last five years [7,10,14], which was confirmed by discussion with different stakeholders in these regions.

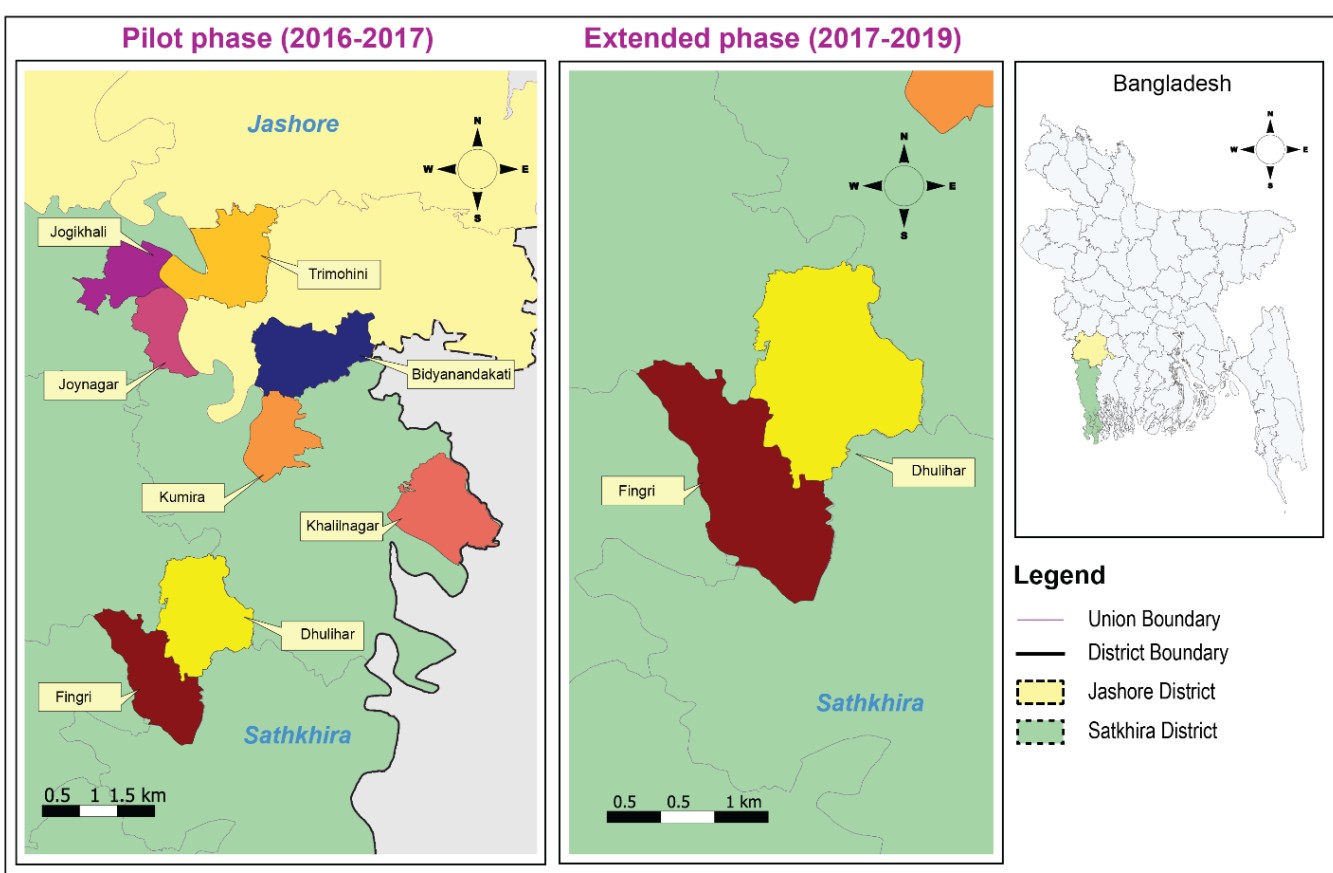

**Figure 1.** Phase–wise Upazilas of the selected Districts for this study.

**Table 1.** The selected study Unions in the pilot phase for the baseline survey (2016) and research work (2017), and in the extended phase for baseline survey (2017) and research work (2018–2019).

| Pilot Phase Areas: Baseline Survey (2016) and Research Work (2017) | | |
| --- | --- | --- |
| District | Upazila | Union |
| Satkhira | Satkhira Sadar | Fingri |
| | | Dhulihar |
| | Tala | Kumira |
| | | Khalilnagar |
| | Kolaroa | Joynagar |
| | | Jugikhali |
| Jashore | Keshobpur | Trimohoni |
| | | Bidyanandakati |
| Extended phase areas: baseline survey—2017 and research work—2018–2019 | | |
| District | Upazila | Union |
| Satkhira | Satkhira Sadar | Fingri |
| | | Dhulihar |

## 2.2. Study Participants and Profile

During both phases, landless marginal farmers having similarly-sized houses (homesteads) were selected from among the affected unions in different districts (Figure 2). All of them were smallholding landless marginal farmers who owned a homestead area ranging from 20 to 30 decimals with no cultivable land. Agriculture was their main occupation, and all were engaged in sharecropping, mainly for rice cultivation, from which they barely managed to attain a 4 to 6-month supply of staple food for their families. To support their families, they worked as agricultural wage laborers. Their homestead-based production of vegetables, fruit, poultry, and livestock was severely hampered because of long-term inundation, i.e., waterlogging. During the waterlogged period, they had to sell live assets such as poultry and livestock because of lack of dry places in the homestead. Indigenous fruit trees were completely destroyed because of inundation for long periods. Thus, their available limited assets were seriously affected and they had barely any income for survival, which consequently exacerbated their living conditions, creating chronic food insecurity.

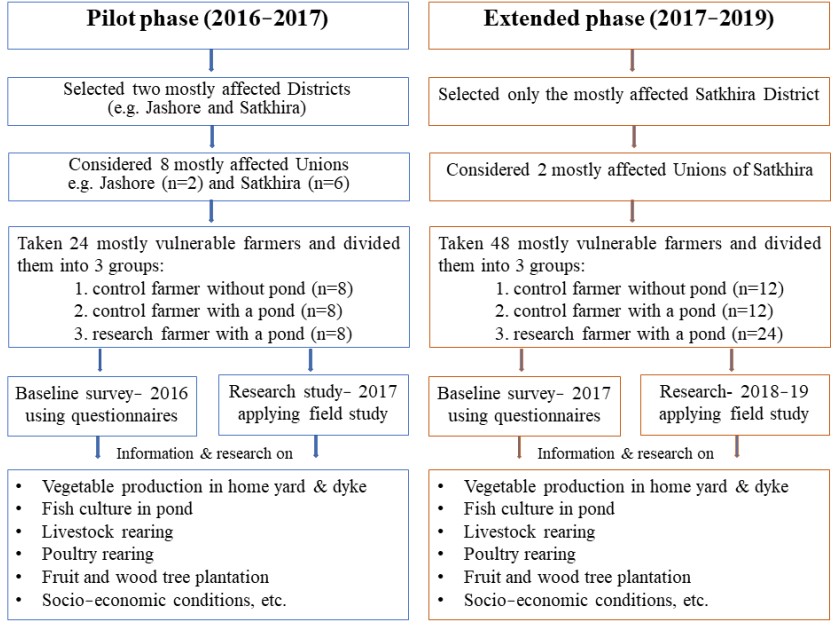

**Figure 2.** Phase–wise study design showing time frame, sample size (n), and data collection for this study.

*2.3. Research Design*

Pilot phase: the pilot study was conducted from March 2016 to February 2017. In this phase, one selected farmer's house had a homestead pond, while the other two houses had no pond, of which one was selected as a research farm (Figure 2). The selected houses were then divided into three groups: (i) control farmers without ponds (eight homesteads); (ii) control farmers with ponds (eight homesteads); and (iii) research farmers (eight homesteads).

Extended phase: the extended study was conducted from March 2017 to February 2019. A total of 48 marginal farmers' houses in the two most affected unions were selected (Figure 2). Among these, twelve houses had homestead ponds. The remaining 36 farmers' homesteads had no pond; of these, 24 farmers' homesteads were selected for the research trial. Thus, the selected houses were divided into three groups: (i) control farmers without ponds (12 farmers); (ii) control farmers with ponds (12 farmers); and (iii) research farmers (24 homesteads).

The selected research homesteads were converted into "variedly elevated homesteads" as recommended by NIRAPAD [21]. This approach is a new idea for coping with waterlogging. To convert a homestead into a variedly elevated homestead, one part of the homestead is raised above flood level by digging out the soil from another part of the homestead (a comparatively lower part with a small ditch). The conversion process for the research homesteads was carried out under the technical guidance of the researchers. Thus, the research homesteads now had ponds with raised dikes and raised land in order to test the new production model of "land- and water-based production technology through a variedly elevated homestead approach".

After the conversion process of the research homestead was finished, the researchers supported the research farmers in developing the layout for production. This production layout included all possible interventions (fish culture in the pond, and fruit, vegetables, poultry, and livestock in the raised part) and adaptive techniques in the homestead in order to maximize the production by acquiring the benefits of both the water body and the land area.

The research farmers were fully guided and economically supported in converting the homestead by digging out the soil for a pond and raising the pond dikes along with other parts of the homestead (Figure 3a,b). In addition, the research farmers received advice on adaptive and alternative agricultural production techniques. After their homesteads were reconstructed according to the design, the researchers provided the following sustainable adaptive and alternative agricultural production technologies:

– Vegetable production using both horizontal and vertical spaces through different resilient techniques using locally available materials (tower/bag/pit/hanging, etc.);
– Fruit and vegetable cultivation using the top and slopes of the pond bank;
– Fish mono- and polyculture in the pond;
– Use of pond surface for vegetables through trellis on the pond;
– Poultry rearing in improved and raised poultry shed;
– Livestock rearing in improved and raised shed.

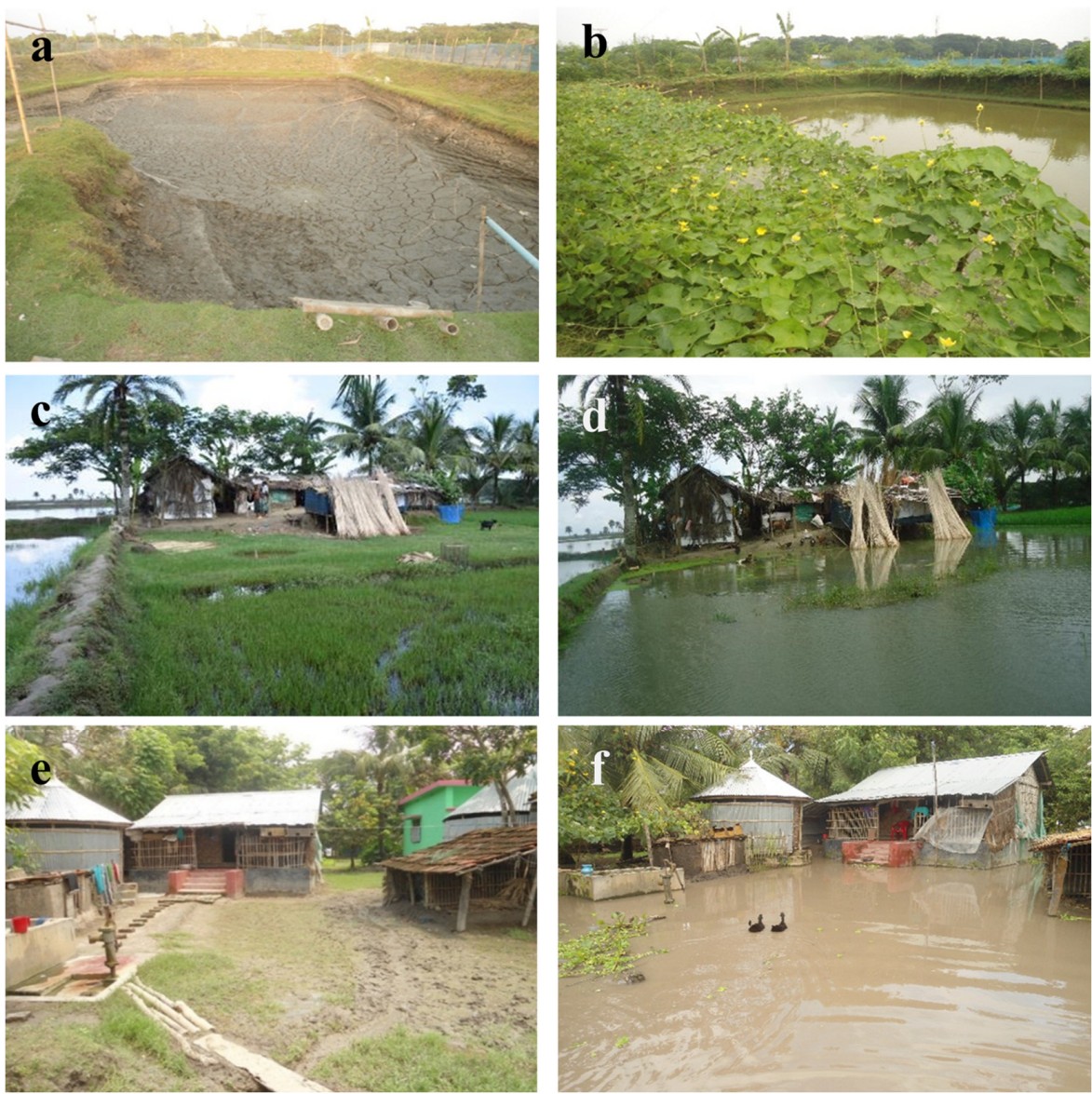

**Figure 3.** Pictures showing a research farmer's household during (**a**) summer and (**b**) rainy season, a control farmer's household with a pond in (**c**) summer and (**d**) rainy season, and a control farmer's household without a pond during (**e**) summer and (**f**) rainy season.

In contrast, none of the control farmers received any support and no change was carried out in their homesteads (Figure 3c–f). They were only asked to provide baseline data and other seasonal production data.

*2.4. Adaptive and Alternative Agriculture Technologies*

After the research homesteads were reconstructed according to the design, adaptive and alternative agricultural production technologies were provided by the researchers. Different vegetable production methods, such as multiple beds, earthen pits, hanging pots, shelf culture, raised beds in the home yard, concrete ring bed, sac bed, tower bed, raised earthen piles, roof top, only vertical culture, combined vertical and horizontal culture, high floor culture, culture in home yard/open space, and fallow isle (following the methods of [22–28]), were followed in order to produce a greater amount of vegetables using homestead land resources. In the pond dikes, around 15–18 types of vegetable (e.g., pumpkins, bitter gourds, long beans, okra, green papaw, and chilies) and 7–10 types of fruit (e.g., bau kul, apple guava, banana, citrus lemon) were cultivated to produce extra crops.

In the case of aquaculture, carp polyculture (a suitable composition of silver carp, catla, rohu, common carp, mrigal, sarputi, and grass carp suggested by previous studies [29–31]), mixed culture of freshwater giant prawns with carp polyculture (e.g., [32–34]), and Asian stinging catfish or shingi monoculture (e.g., [35–37]) were practiced in order to produce a greater amount of fish while following sustainable environment-friendly aquaculture techniques.

Research farmers were advised to plant different fruit trees on the pond dike and homestead periphery in order to grow fresh fruits for year-round consumption. They were told to plant saplings of guava, lemon, papaya, jujube, sapota, mango, kamranga, pomegranate, rose apple, etc., as these plants do not usually damage pond dikes and can support integrated aquaculture. The research farmers were advised to construct raised sheds in the newly raised plinth area in order to avoid inundation by floods and waterlogging. It was suggested that they raise the plinths by digging/re-excavating their homestead enclosed derelict mini-pond to allow chickens room to move back and forth when searching for food in the open dry courtyard. The farmers were advised to keep/store sufficient dry balanced food to feed their chickens during the waterlogged season. Rearing chickens and ducks together is a traditional primitive method in rural areas of Bangladesh. Therefore, research farmers were advised to make resilient high sheds made of wood, bamboo, or brick and to arrange 2–3 separate chambers/shelves within the shed. It was suggested that they keep ducks in one chamber and chickens in another. In the case of a three-chamber shed, it was suggested that they keep chicks or ducklings in the third chamber. The farmers were strongly advised to store sufficient supplementary food for feeding their rearing chickens and ducks during the rainy season, when chickens have very limited room to search for food while grazing.

Livestock (cows, buffalo, goats, and sheep) are highly vulnerable to floods and waterlogging, and it is very hard to rear them, especially in disaster-prone areas. During flooding and waterlogged periods, both the homestead and the grazing field for cattle are inundated, and they have very limited or no room to graze in the field. Moreover, cattle suffer from lack of suitable living sheds due to prolonged waterlogging. Due to these circumstances, the research farmers were provided with support for building waterlogging-resilient cattle sheds in their homesteads in such a way that flood water would not be able to reach the floor of the shed. They were told to store sufficient dry food for their cattle in order to feed them during flooding and waterlogged periods.

### 2.5. Data Collection and Analysis

Data were collected by the field assistants using a questionnaire (Supplementary Materials) validated by a committee of experts from different fields (agriculture, fisheries, agricultural economics, forestry, disaster management, livestock, and social sciences). In both phases, the collected data provided information about existing assets (productive assets such as land resources, ponds, livestock, poultry, and trees), present income sources, daily and monthly average income and expenditures, etc. While the adaptive and alternative technologies were only applied to the selected research farms, monthly monitoring was carried out on all farms in order to provide inputs and collect information. Finally, the data were rearranged, compiled, and analyzed with appropriate statistical models using R software version 4.0.5 [38]. The descriptive statistics (means and SEs) were calculated in R using the 'psych' package. The Shapiro–Wilk test for normality and Levene's test for homogeneity of variance were carried out with the 'onewaytests' package. Because most of the response variables were 'count data' (income in money), which does not comply with the assumptions of any parametric model, the no-linear Poisson regression model was used, as this model is usually suggested for count and percent data. The 'quasi-Poisson' regression model was applied here using the 'pscl' package, which is useful as it has a variable dispersion parameter to minimize over-dispersion of data [39]. In this model, 'income from different resources' was included as the 'response variable', while 'farmer group' and 'season' were incorporated as 'fixed factors'. Tukey's post hoc tests were subsequently

carried out for pair–wise comparison using the 'multcompView' and 'emmeans' packages. All graphs were made with the 'ggplot2' package.

## 3. Results

### 3.1. Income from Homestead Vegetables

Our analysis revealed significant variation among farmers because of their income from homestead vegetable production in both the pilot phase ($p < 0.001$, Figure 4a) and the extended phase ($p < 0.001$, Figure 4b). The subsequent post hoc tests revealed that research farmers' income from vegetable production was significantly higher than that of the other two groups of farmers ($p < 0.001$), while no significant variation was found between control farmers with ponds and those without ponds during either the pilot (Figure 4a) or the extended phase (Figure 4b).

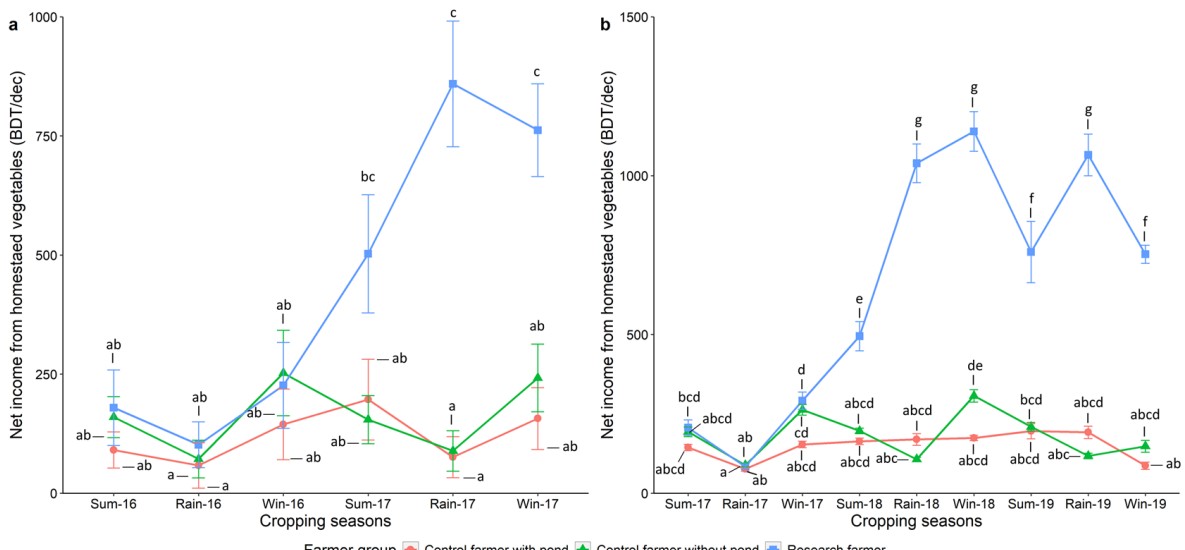

**Figure 4.** Income (BDT/dec) of the selected farmers from homestead vegetables during different seasons and phases. Findings are shown in (**a**) the pilot phase and (**b**) the extended phase. Here, 1 USD = 105.00 BDT. Values are presented as mean ± standard error (SE). Differences in small letters presented above or near each error bar indicate significant differences in income between different farmers ($p < 0.05$).

The findings showed significant differences in income due to seasonal variations in both phases ($p < 0.001$). It was noticed that the research farmers' income from vegetable production increased significantly compared to other farmers after receiving assistance from the project in both 2017 (pilot phase) and in 2018–2019 (extended phase) in comparison to the non-project years of 2016 (pilot phase) and 2017 (extended phase). On the other hand, the study showed no significant variation between the incomes of the other groups of farmers (Figure 4a,b).

### 3.2. Income from Fish

Fish production, or aquaculture, is one of the most promising sectors in this region. Our data showed that the income from this sector significantly varied among the farmers during both phases ($p < 0.001$), which mainly indicates the significantly higher income of both the research farmers and the control farmers with ponds compared to the control farmers without ponds. Post hoc tests revealed no significant variation between the income of research farmers and control farmers with ponds in either the pilot ($p = 0.69$) or extended ($p = 0.66$) phases.

The study revealed that this fish income significantly depended on seasonal variation in both phases ($p < 0.001$). In 2016 (baseline survey), the research farmers' income from this sector was significantly lower than that of the control farmers with ponds ($p < 0.001$,

Figure 5a), which sharply and significantly increased ($p < 0.001$, Figure 5a) during 2017 after the farmers received technical support from this project. The same scenario was observed in 2017 (baseline survey); the adaptive technologically supported farmers raised their income significantly in 2018 and 2019 compared to the control farmers with ponds ($p < 0.001$, Figure 5b).

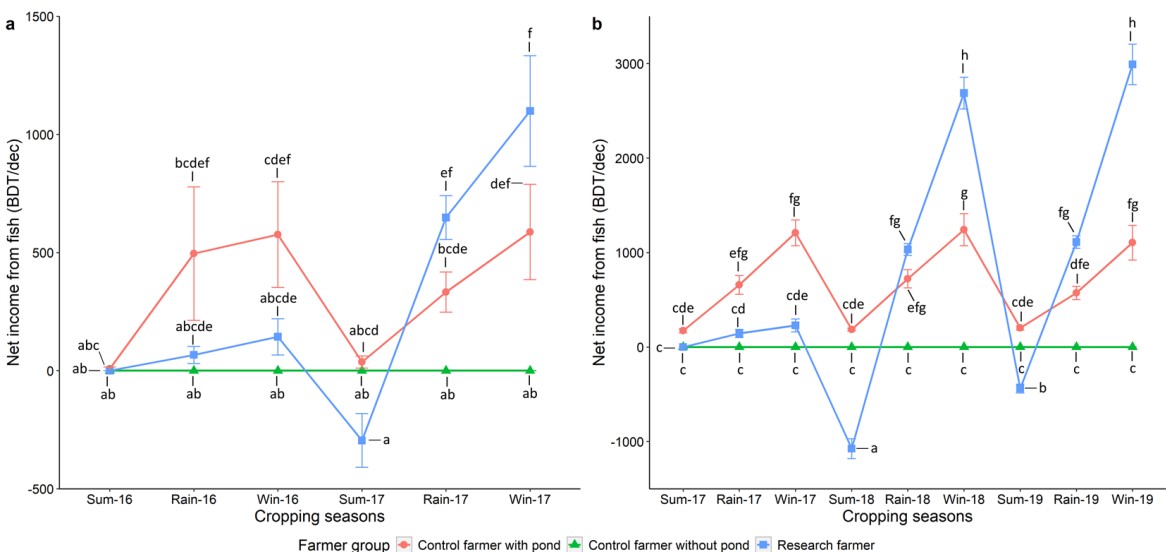

**Figure 5.** Income (BDT/dec) of the selected farmers from fish production and sale during different seasons and phases. Findings are shown in (**a**) the pilot phase and (**b**) the extended phase. Here, 1 USD = 105.00 BDT. Values are presented as mean ± standard error (SE). Differences in small letters presented above or near each error bar indicate significant differences in income between different farmers ($p < 0.05$).

### *3.3. Income from Dike Cropping*

In dike cropping, no significant variation in income was found between research farmers and control farmers with ponds ($p = 0.57$, Figure 6a) during 2016; however, these results changed in 2017, when research farmers earned significantly higher income than control farmers with ponds ($p < 0.001$, Figure 6a). According to the baseline survey taken in 2017, the research farmers' dike cropping income was significantly lower than that of the control farmers with ponds ($p < 0.001$, Figure 6b); this position was abruptly reversed in 2018 and 2019, when research farmers significantly enhanced their income from dike crops ($p < 0.001$, Figure 6b) with the assistance of project personnel.

The analysis unveiled no significant variation in dike crop income among the seasons of 2016 (Figure 6a); while a marginally significant variation was observed between summer 2017 and rainy season 2017 ($p < 0.05$) during the pilot phase, no significant variations were found among the rest of the seasons (Figure 6a). In 2017 (extended phase), no significant variation was found amongst the various seasons in terms of dike crop income (Figure 6b). Presumably, it was the rainy seasons of 2018 and 2019 and the winter of 2018, rather than the summer seasons, that were the best dike crop income seasons ($p < 0.001$, Figure 6b).

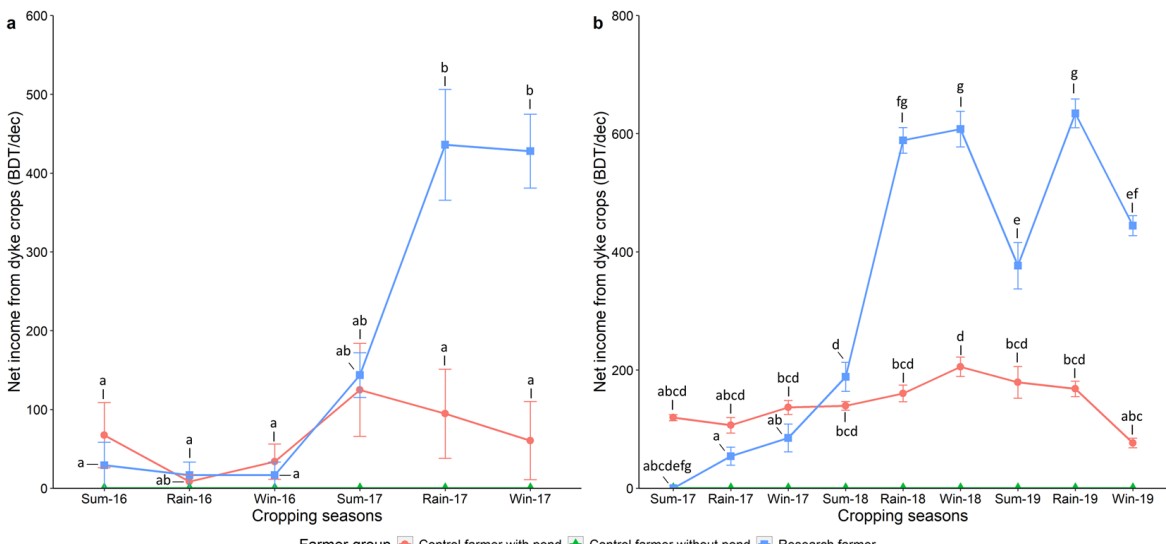

**Figure 6.** Income (BDT/dec) of the selected farmers from pond dike cropping during different seasons and phases. Findings are shown in (**a**) the pilot phase and (**b**) the extended phase. Here, 1 USD = 105.00 BDT. Control farmers without a pond had no dike with which to produce crops. Values are presented as mean ± standard error (SE). Differences in small letters presented above or near each error bar indicate significant differences in income between different farmers ($p < 0.05$).

### 3.4. Income from Livestock

The research farmers' income from livestock was significantly higher ($p < 0.01$) than that of the control farmers with ponds, while no significant variation was found among the other groups during the pilot phase (Figure 7a). In the extended phase, the research farmers' income was significantly higher than both groups of control farmers ($p < 0.001$, Figure 7b), while control farmers without ponds earned significantly more from livestock than control farmers with ponds ($p < 0.05$, Figure 7b).

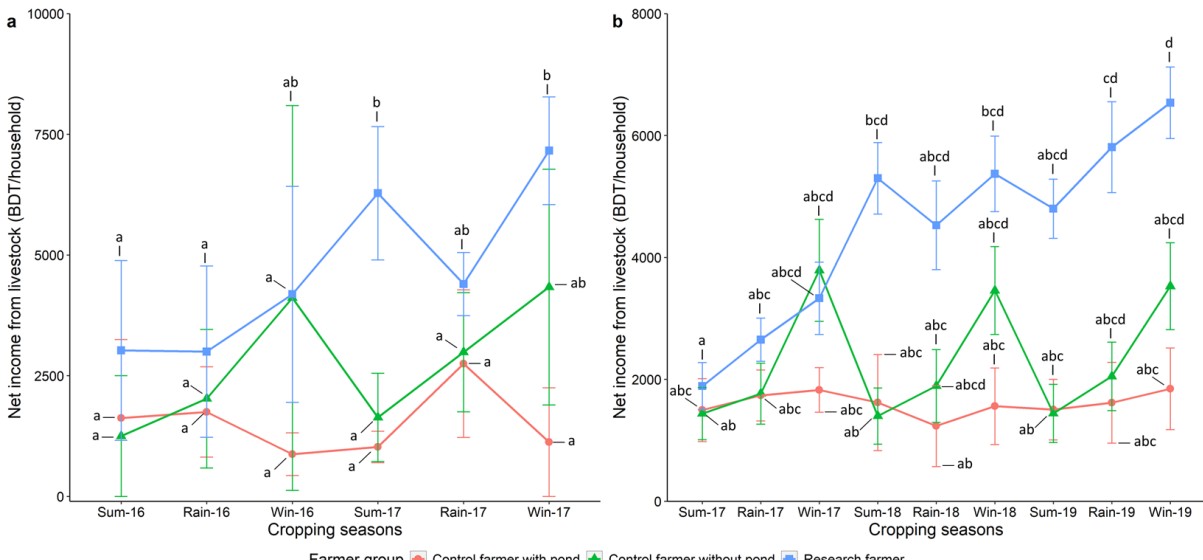

**Figure 7.** Income (BDT/dec) of the selected farmers from livestock during different seasons and phases. Findings are shown in (**a**) the pilot phase and (**b**) the extended phase. Here, 1 USD = 105.00. Values are presented as mean ± standard error (SE). Differences in small letters presented above or near each error bar indicate significant differences in income between different farmers ($p < 0.05$).

The study revealed no significant differences caused by seasonal variations during the pilot phase ($p = 0.67$, Figure 7a), though significant variations were found during the extended phase ($p < 0.001$, Figure 7b). Post hoc analysis showed that incomes were significantly different between summer 2017 and winter 2017 when the baseline survey was carried out ($p < 0.01$, Figure 7b). The findings showed that the research farmers' income increased significantly with support from the project, which ultimately enhanced their income after summer 2018 ($p < 0.001$, Figure 7b).

### 3.5. Income from Poultry

During the pilot phase, the control farmers with ponds earned a significantly higher amount of money from poultry than the control farmers without ponds ($p < 0.05$); otherwise, no significant variation was observed among different farmers' poultry income (Figure 8a). In the extended phase, research farmers' income from the poultry sector was significantly higher ($p < 0.05$, Figure 8b) than that of both groups of control farmers, while control farmers with ponds had significantly higher income than control farmers without ponds ($p < 0.01$, Figure 8b).

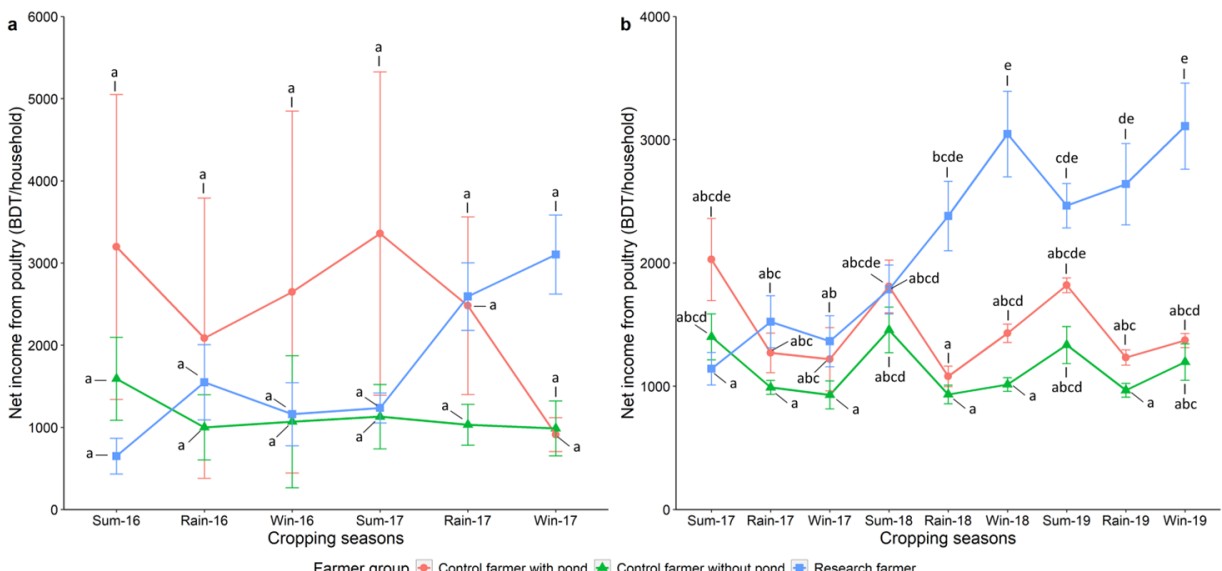

**Figure 8.** Income (BDT/dec) of the selected farmers from poultry during different seasons and phases. Findings are shown in (**a**) the pilot phase and (**b**) the extended phase. Here, 1 USD = 105.00 BDT. Values are presented as mean ± standard error (SE). Differences in small letters presented above or near each error bar indicate significant differences in income between different farmers ($p < 0.05$).

The study revealed no significant variation among seasonal incomes from poultry in the pilot phase ($p = 0.98$, Figure 8a). On the other hand, the research farmers' income during the extended phase sharply increased after receiving funds to raise poultry (from the 2018 rainy season onwards) compared to both groups of control farmers ($p < 0.01$, Figure 8b).

### 3.6. Status of Different Trees

In the pilot phase, the number of forest trees on research farmers' lands significantly increased after the project was launched in 2017 ($p < 0.05$, Figure 9a), while the number of fruit trees did not significantly vary among the different groups of farmers ($p = 0.6$). During the extended phase, the total numbers of both tree types significantly increased on research farmers' lands compared to both groups of control farmers ($p < 0.001$, Figure 9b), while no significant variation was found between the two groups of control farmers.

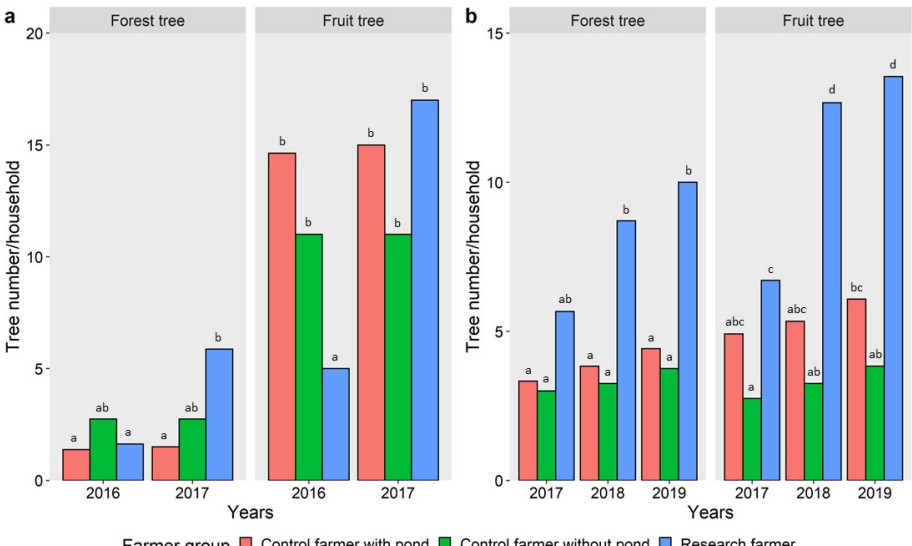

**Figure 9.** Average number of trees on the selected farmers' homesteads during different years and phases during (**a**) the pilot phase and (**b**) the extended phase. Values are presented as the average numbers of different types of trees. Differences in small letters presented above each bar indicate significant differences in the number of trees between different farmers ($p < 0.05$).

### 3.7. Net Income from Homestead Based on All Farming Systems

The data on the overall net income from all crops in 2016 (vegetables, aquaculture, dike crops, livestock, and poultry) revealed no significant variation among the groups of farmers ($p = 0.48$) or across the different seasons ($p = 0.42$, Figure 10a). However, after the pilot phase intervention was made in 2017, the total income from all crops for the research farmers boomed, increasing significantly compared to both groups of control farmers ($p < 0.01$), which drastically affected seasonal income during the rainy season and winter of 2017 ($p < 0.01$, Figure 10a).

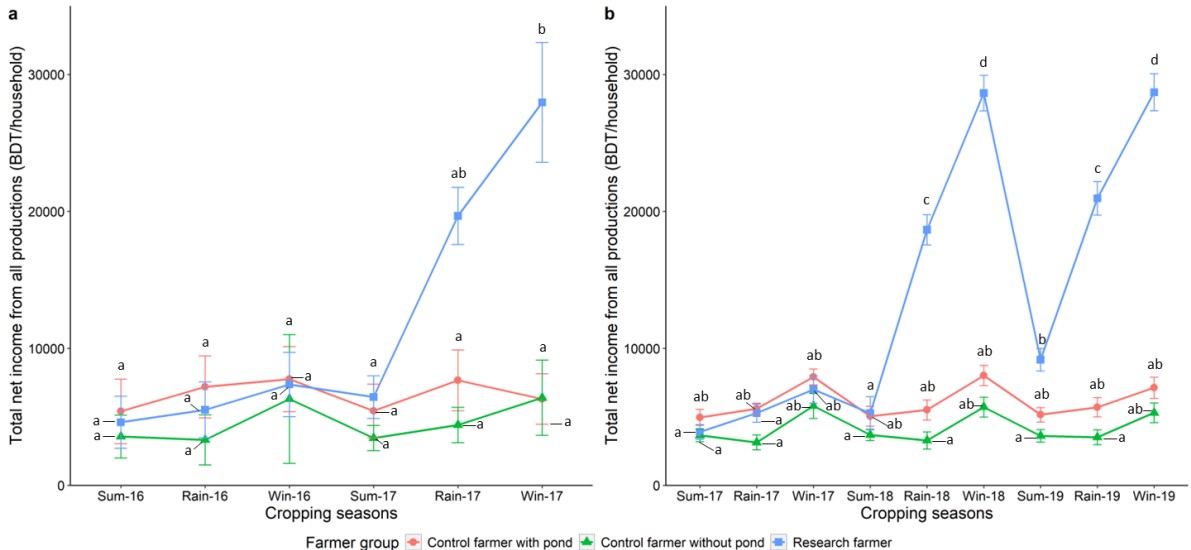

**Figure 10.** Average net income (BDT/household) of the selected farmers from all farming systems during different seasons and phases. Findings are shown in (**a**) the pilot phase and (**b**) the extended phase. Here, 1 USD = 105.00 BDT. Values are presented as mean ± standard error (SE). Differences in small letters presented above or near each error bar indicate significant differences in income between different farmers ($p < 0.05$).

During the extended phase, the overall net income from all crops was not significantly different among the groups of farmers (Figure 10b). This condition then changed, with the research farmers earning a significantly higher amount of money than either group of control farmers ($p < 0.001$, Figure 10b). The research farmers' huge increase in income ultimately prompted significant seasonal variation, with even the worst rainy season and winter season becoming highly productive seasons ($p < 0.001$, Figure 10b).

### 3.8. Research Farmers' Crop–Wise Seasonal Net Income

This study clearly shows that the overall income of research farmers increased significantly during the rainy and winter seasons in both phases, which were not as productive before the implementation of this project (Figure 11a,b).

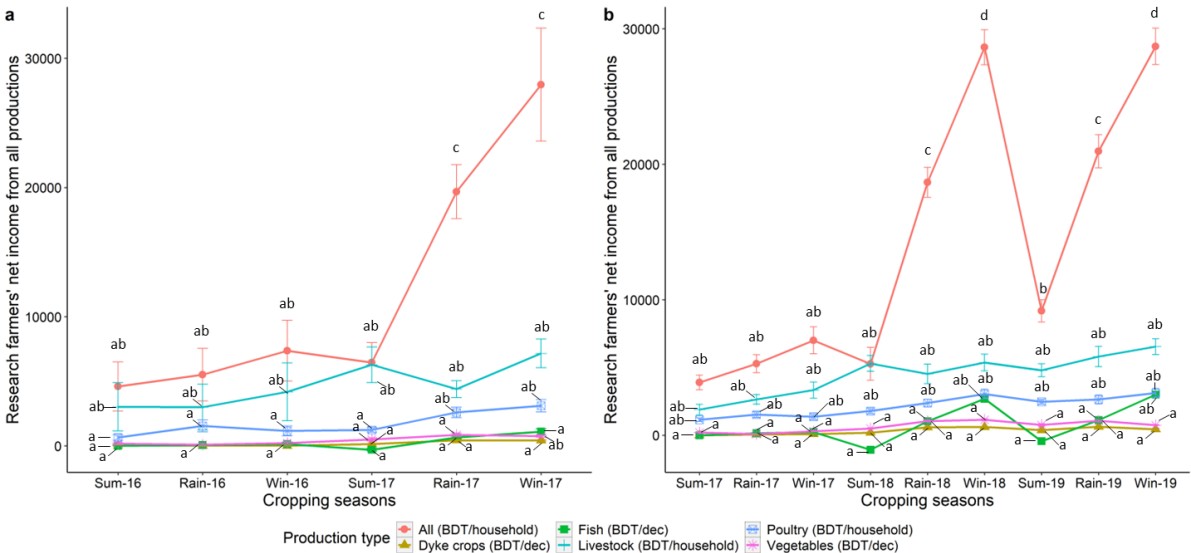

**Figure 11.** Crop–wise seasonal net income from the homestead farming of research farmers during different seasons and phases. Findings are shown in (**a**) the pilot phase and (**b**) the extended phase. Here, 1 USD = 105.00 BDT. Values are presented as mean ± standard error (SE). Differences in small letters presented above each error bar indicate significant differences in total net income between different crops in different seasons ($p < 0.05$).

### 3.9. Utilization of Homestead Land during Rainy Season

Figure 12a,b depicts the land use pattern of different farmers, in particular during the worst rainy season, when their homestead land became waterlogged for a prolonged period and was unavailable to use for livelihood purposes. Prior to the two study phases (the survey periods of 2016 and 2017), the selected research farmers did not know how to use their limited land resources to produce certain valuable agricultural products by applying certain common tools and techniques. Following the initiation of this project (the two research periods, that is, the 2017 pilot phase and the 2018–2019 extended phase), research farmers gained the opportunity to learn how to maximize their available homestead lands (Figure 12a,b) and produce valuable resources in order to support their livelihoods. In both figures, it is clearly illustrated that, with this project's assistance, research farmers were able to maximize their limited unused lands by turning them into productive resources.

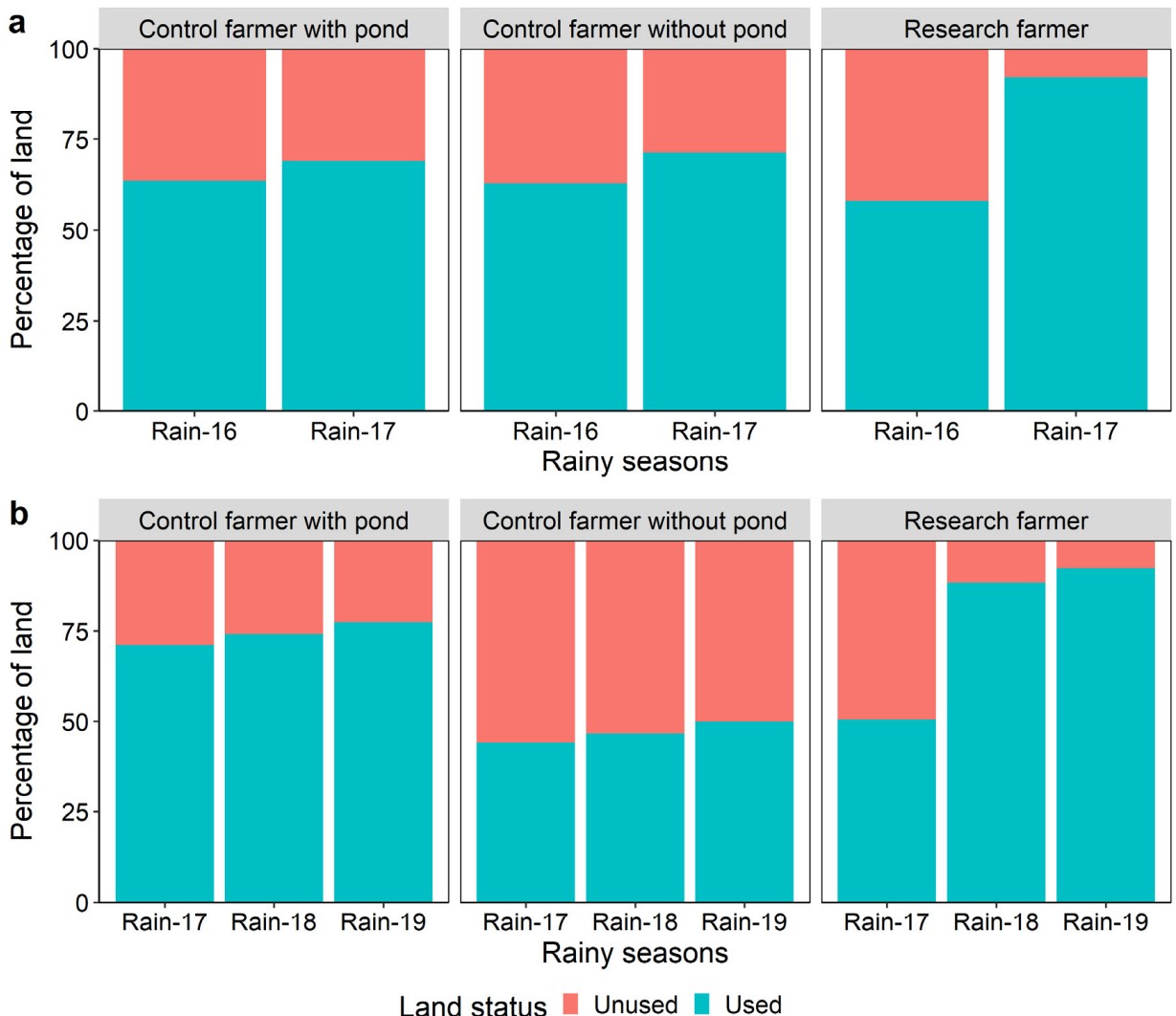

**Figure 12.** Uses of homestead land by different farmers especially during rainy season. Findings are shown for (**a**) the pilot phase and (**b**) the extended phase.

### 3.10. Creating Job Opportunities

One of the major contributions of this project was to create jobs, and especially to empower women to earn money to cover their family expenses. The analysis explicitly exposed that, prior to the project, men were the only earning members of farming families, and that their number was not increased significantly by the project (Figure 13a,b). Surprisingly, the quantity of female earning members increased significantly in almost every research farmer's household (Figure 13a,b). This is one of the most important outcomes of this project, as it increases overall income by using limited resources to cope with adverse waterlogging through adaptive technological devices.

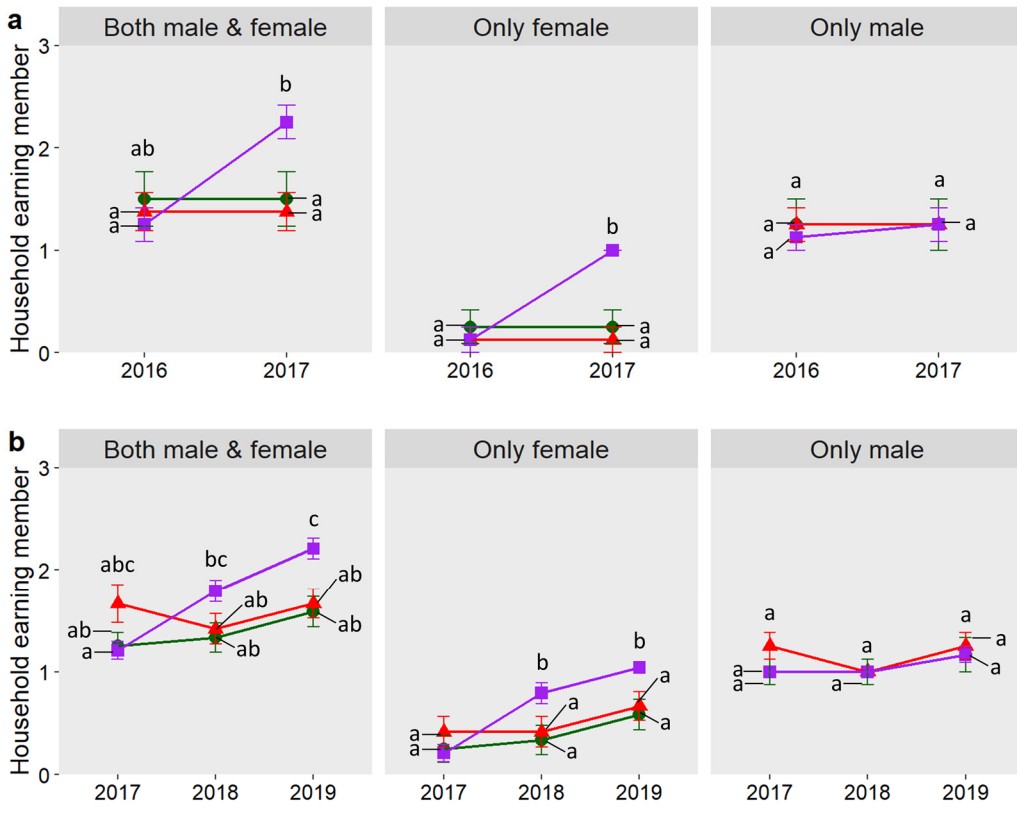

**Figure 13.** Number of earning members in each farming group during different phases. Findings are shown for (**a**) the pilot phase and (**b**) the extended phase. Values are presented as mean ± standard error (SE). Differences in small letters presented above or near each error bar indicate significant differences in number of earning members between different farming families ($p < 0.05$).

## 4. Discussion

The study data demonstrate that the researched adaptive and alternative technologies significantly enhanced the overall crop production of every research farm, particularly in homestead vegetable production, pond dike cropping, and aquaculture farming systems, enabling the farmers to successfully cope with waterlogging. Homestead vegetable production can play a significant role in coping with waterlogging, and can improve food security for resource-poor rural households in developing countries such as Bangladesh [16,28,40]. In a previous study, Adri and Islam [16] showed how waterlogged communities of the Keshabpur upazila in Jessore district, Bangladesh could maintain coping strategies through agricultural cropping. The affected people prepared seed beds by partially raising a piece of land with soil/mud, allowing them to cultivate winter crops ('rabi'). In a review, Rao and Li [41] recommended soil management practices such as ridging and furrowing and the creation of raised beds before any prolonged waterlogging or flooding in order to avoid severe damage to crop production. Ring gardening, floating vegetation, grafting, etc., were utilized as new farming technologies by local farmers to cope with waterlogging [16,28,40]. The present study used adaptive technologies to support the research farmers in improved and sustainable production of homestead vegetables in order to help them cope with prolonged waterlogging.

Asaduzzaman et al. [42] conducted a study to assess the costs and benefits of different types of homestead vegetable gardening in terms of improving household food and nutritional security in rural Bangladesh (34 upazilas in five districts). They found that, on average, 2.5 types of vegetables per household were grown out of 21 species of traditionally cultivated vegetables. Among these, the bottle gourd and hyacinth bean were grown by

more than 90% of households under this gardening system. On the other hand, 17 types of vegetables were produced per household during vegetable gardening developed with year-round fixed plots/beds. In the present study, around 15–18 types of vegetables were suggested for cultivation, which yielded a huge amount of vegetables; thus, the findings of this research seem to corroborate previous studies. Hasan and Sultana [43] conducted a study in the flood-affected 'char' land, and found that most households had an average homestead vegetable farm of 6.71 decimals, where they cultivated spinach, bitter gourds, cowpeas, pumpkins, okra, water spinach, and pointed gourds in the summer season and red amaranth, brinjal, tomatoes, beans, radishes, and peppers in the winter season. A considerable portion of these vegetables were consumed by the farmers themselves, which provided them with food security; the rest were sold, which provided economic security and socio-economic recognition. The majority of the homestead vegetables produced by the research farmers in the present study were consumed by the farmers and their families, with the remaining portion sold to the market. Thus, this farming system provides dietary diversity and food security as well as economic benefits, allowing farmers to better cope with waterlogging problems. Similar approaches to the production of various types of homestead vegetables have been followed by other studies [10,28,40]; these were applied in the present study to develop a sustainable homestead vegetable production system that can provide support and resilience in the waterlogged areas of Bangladesh.

Integrated fish production or aquaculture practices are one of the most significant activities in terms of coping with waterlogging and resulting improvements to the livelihoods of affected vulnerable people [44–46]. Therefore, the present study provided financial and technical support to the research farmers for production of different kinds of fish using their household ponds. Although at the beginning of fish production (summer season in both phases) the research farmers saw a reduction in their net income because of investment in pond preparation and purchase of seed fish and feed, their income was significantly increased (Figure 5a,b) after they received technical support from this project. Bloomer [47] found that carp was the most commonly cultured fish species in the household ponds of the waterlogged Satkhira and Khulna districts, with polyculture mostly being practiced. This provided a comparatively higher income for households with ponds compared to households without ponds, and households with ponds were self-sufficient for a longer period of time than households without ponds [47]. Hasan [48] conducted another study in the waterlogged area of Noakhalia, Bangladesh, where he found that traditional carp polyculture was very popular and that polyculture of carp with prawns did not significantly increase the overall costs of aquaculture. Moreover, local farmers strongly preferred prawns, which already have a very good market. Therefore, the research farmers in the present study were also advised to farm a mixed culture of carp and prawns for higher profits. In Bangladesh, a mixed culture of carp, tilapia, and prawns has been shown to provide farmers with higher net profit than monoculture of either carp or tilapia [33,49]. Therefore, the present study suggests utilizing a polyculture or mixed culture of different fish in order to produce a higher yield of finfish and prawns using the same pond resources. In areas where the research farmers had unused derelict ponds, a monoculture of Asian stinging catfish (shingi) was recommended, as this carnivorous fish can eat other co-culture species; shingi monoculture has already shown a higher weight gain, better feed conversion ratio, and higher net profit in various studies [36,37,50]. In the present study, a monoculture of Asian stinging catfish at a recommended stocking density of 100,000 individuals/ha was suggested, and this provided good financial support for research farmers using unused derelict ponds during waterlogging.

Previous studies have suggested adaptive production techniques such as floating bed vegetable cultivation and dike cropping (mainly for vegetables) in order to cope with changing climatic conditions (e.g., floods, waterlogging); this can eventually reduce vulnerabilities and increase food security [51–53]. Vegetable cultivation on pond dikes is an improved agricultural practice that ensures maximum use of a small household or farm. In their study, Islam et al. [53] showed that cultivation of different vegetables (e.g., pumpkins,

cucumbers, bitter gourds, teasle gourds, broad beans) on pond dikes could enable higher profits from this farming system. Azad et al. [54] conducted a study in coastal areas of Bangladesh, finding that many farmers grew vegetables on the dikes of their ponds (ghers). They cultivated various types of vegetables (e.g., pumpkins, bitter gourds, long beans, okra) using trellises above the trench in order to enhance yields. Azad et al. [50] explained that a number of farmers grew banana plants on gher dikes, and that banana leaves were used as feed for grass carp. In another study, Adri and Islam [16] showed that the levees (dikes) of the fishing ponds in areas in the waterlogged Keshabpur upazila in Jessore district were raised up to a certain level as a precautionary and safety measure, ensuring that fish could not leave the ponds; certain kinds of crops were cultivated on these dikes. The research farmers in the present study adopted similar kinds of culture practices, using their pond dikes to enhance their overall crop production and cope with waterlogging.

Natural disasters such as heavy rainfall, floods, and waterlogging can damage crops and severely affect livestock, poultry, and settlements [55–57]. Several studies have suggested rearing livestock and/or poultry as suitable strategies for affected communities to generate alternative livelihood or income sources in order to cope with waterlogging [56,58,59]. During waterlogging, farmers are advised to keep their livestock on raised platforms [60,61]. Consistent with the above studies, the present study suggested that the research farmers rear livestock (cattle, goats, and poultry); the study provided financial support for the farmers to buy livestock and feed, built raised platforms for their shelter, and provided technical advice on proper rearing. As a result, the research farmers obtained the highest profit from this sector.

Prolonged floods and waterlogging can severely affect growth and even cause mortality for many tree species [62,63]. Studies have recommended planting selected tree species that can adapt to the changing conditions [64,65]. For example, certain agroforestry and fruit trees are strongly suggested for planting on homestead lands and/or pond dikes, as these trees can withstand prolonged floods and waterlogging, minimize soil erosion, enhance soil fertility, diversify crop yields, contribute to financial support, and increase the resilience of the household's livelihood [66–71]. The research farmers in the present study were advised and trained to carry out agroforestry on their lands, allowing them to achieve effective, sustainable, and cost-efficient adaptation to waterlogging.

The present study shows that research farmers learned from this project how to maximize their homestead land resources for various agricultural production. Adri and Islam [16] revealed similar findings; the affected farmers in their study raised part of their homesteads or agricultural lands to cultivate different crops, rear cattle, etc., while the lower part was excavated to create a strong dike in order to save fish and produce dike crops. Thus, the farmers ensured that they were using most of their land resources through different types of agricultural farming. Awal [10] demonstrated how affected people practiced different locally adaptive agricultural farming systems, using most of their available land resources, in order to cope with waterlogging. Another study by Asaduzzaman et al. [42] showed that farmers who utilized their homestead land resources for improved gardening could ensure better food security than traditional gardeners. The same study revealed that the likelihood of food security was increased by a factor of 4.52 when women were engaged in homestead gardening in addition to male gardeners; this finding supports those of the present study, in which women on the research farms were empowered through this project to ensure food security for their family. Women in flood-affected areas are mostly vulnerable and face difficulties in finding adequate shelter, food, safe water, and fuel for cooking, as well as problems with maintaining personal hygiene and sanitation; all of this prevents women from performing their usual roles at home [60,72–74]. These problems are related to women's gender identity and social roles. Many poor and destitute women remain unemployed during and after floods. In addition, women suffer from domestic violence and are subject to harassment when taking shelter or refuge at community centers. These particular vulnerabilities and problems interrupt women's mitigation efforts and adaptation capacities in disaster risk reduction. Fortunately, the present study achieved

great success by empowering at least one woman in each research farm group, thereby saving them from the various difficulties mentioned above.

## 5. Conclusions and Recommendations

The present study has documented how to improve the living conditions of waterlogged affected people, provide shelter for their domestic animals, and enhance the production of homestead vegetables, pond dike crops, fish, agroforestry, livestock, and poultry. The study implemented an integrated agriculture farming approach in order to ensure food security through sustainable production of year-round micronutrient-rich plant and animal diets at the household level. Although the research achieved success as a model study, there were limitations during project implementation, such as the very limited number of farmer groups, scarcity of funds, the short time period of the study, the small scale of the project, transportation problems (especially during the rainy season), lack of collaboration and co-operation in certain cases, etc. However, the successful findings of this model study can be used to help other farmers with similar waterlogging and long-term flood problems. Finally, to promote adaptations in the agricultural sector, increase sustainable productivity, and reduce waterlogging-induced loss and damage under different farming systems, the following recommendations are made, which can be helpful for other affected communities with similar contexts:

- More long-term studies are needed to investigate physical, biological, and socio-economic systems with higher vulnerability to waterlogging;
- Adaptive, alternative, and sustainable development policies and strategies are suggested for fostering ecosystem and agriculture-based adaptation;
- Effective community-based and participatory resource management systems should be developed in order to utilize unused land and cope with waterlogging;
- Collaboration and co-operation with local and international research and funding organizations are needed for the implementation of this kind of project;
- Financial instruments (e.g., risk insurance, adaptation clearinghouse, soft loans for affected farmers, etc.) should be strengthened for waterlogging adaptation;
- Capacity-building programs (institutional governance, infrastructure, and human resources) and appropriate training should be provided to help farmers cope with waterlogging through adaptive and alternative agro-farming;
- All agriculture-based adaptation options should be promoted and made available in the most cost-efficient way;
- Adaptive, alternative, and sustainable agro-farming systems should be promoted in order to ensure maximum utilization of resources in waterlogging-affected areas;
- The availability of agricultural farming inputs should be ensured during adverse periods in order to cope with waterlogging quickly and effectively;
- Transportation and communication systems should be built to allow waterlogging-affected people to use them for livelihood purposes;
- As part of future studies, women's socio-economic empowerment and health status should be considered; and finally,
- More research should be carried out to demonstrate different ecosystem-based agricultural adaptations for coping with waterlogging.

**Supplementary Materials:** The following supporting information can be downloaded at: https://www.mdpi.com/article/10.3390/su15032087/s1, Data collection questionnaire for applied research.

**Author Contributions:** Conceptualization, M.M.R., T.K.C., A.A.M. and V.K.; methodology, M.M.R., T.K.C. and A.A.M.; project administration, T.K.C., A.A.M. and V.K.; data curation, M.M.R. and A.A.M.; formal analysis, M.M.R. and A.A.M.; writing—original draft preparation, M.M.R. and T.K.C.; writing—review and editing, M.M.R., T.K.C., A.A.M. and V.K.; visualization, M.M.R.; project supervision, M.M.R., T.K.C. and A.A.M.; funding acquisition, T.K.C., A.A.M. and V.K. All authors have read and agreed to the published version of the manuscript.

**Funding:** This project was supported by funding from European Union Civil Protection and Humanitarian Aid (ECHO) and from Action Against Hunger (ACF), grant number 2016_00040_RQ_01_01, Z-fund.

**Institutional Review Board Statement:** Not applicable.

**Informed Consent Statement:** Not applicable.

**Data Availability Statement:** The data presented in this study are available on request from the first and corresponding authors.

**Acknowledgments:** The authors cordially thank all consultants and particularly Dioula, Action Against Hunger, France to provide technical assistances, and field staffs for their relentless efforts to monitor and collect data regularly. They also are very grateful to Shuvo, for data compilation and arrangement, and Hena, MEAL for preparing maps. The authors are especially very thankful to the Donor which assisted by providing the required funds to carry out this study successfully.

**Conflicts of Interest:** The authors declare no conflict of interest.

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
