# Peer review of "Land- and Water-Based Adaptive Farming Practices to Cope with Waterlogging in Variably Elevated Homesteads"

_sustainability, doi:10.3390/su15032087_

Round 1

Reviewer 1 Report

Sustainability 2144993 Comments

Useful research and well written, tone is a bit sensational and unbecoming of a scientific paper at times but the English language is excellent.  Overall, I think the information provided is quite useful and could be reported in multiple outlets.  However, this document needs to be streamlined to science findings for this outlet.

I think the overall analysis is good and would only recommend removing sensational, or editorializing, language.  An area to expand with the data would be more on the net financials and long term potential of the change in practices.

Abstract is clearly written but not of appropriate scientific tone, seems more written for translation of article to popular press. No specific data was provided in abstract.

Line 56:  I had to look up upazila, but seeing the definition I believe it is appropriate to continue using this term.  As it is used frequently, the meaning becomes clear.

Line 87: dramatic language not needed in scientific paper.  Phrasing as “Considering the conditions of these vulnerable communities…” will be adequate.  The rest of the sentence also becomes confusing with the sensational language.  How can someone solve an unavoidable problem?  Perhaps just say that much research has been done on the problem and then get to literature review.

Literature reporting is inadequate.  This paper states that study have been done but provides no relevant work to indicate how this study fits into the broader research.

Line 104: not clear why these document were collected or useful to the study.

Line 109: What are unions?  Clarify relation of this term to an upazila?

Section 2.2 and 2.3 seem conflicting to me.  2.2 states that one farmer with no pond was chosen while section 2.3 states that three farmers (some with ponds) were chosen.  This needs to be clarified, perhaps it was two aspects of the study.

Method section does not state over what time period data was collected or over what timeframe expected differences would be seen.

I am glad that production and food security increased with intervention, but the cost as compared to benefit is unclear.  Who will provide the cost of developing water retention ponds, vertical infrastructure for vegetables, etc.?  I get the sense that individual landowners could not afford such.  Net income is finally addressed in section 3.7.  However, it should be explored more, perhaps considering the time to benefit of investment in research practices.

Section 3 – why are results in terms of income instead of yield?  Is it common for these farmers to sell all products and purchase own food separately.  I feel like the amount of food kept on hand has been overlooked and therefore family size may be a confounding factor.  Also, how does income compare to expense?  Perhaps there is not a difference in net profit and therefore food stability would be a better metric than income.

Line 364: What data supports the statements made on women’ status?  I think this would be better included as a discussion point on potential benefits of the reported economic data than reporting as data.

Author Response

# Response to reviewer-1

Useful research and well written, tone is a bit sensational and unbecoming of a scientific paper at times but the English language is excellent.  Overall, I think the information provided is quite useful and could be reported in multiple outlets.  However, this document needs to be streamlined to science findings for this outlet.

I think the overall analysis is good and would only recommend removing sensational, or editorializing, language.  An area to expand with the data would be more on the net financials and long term potential of the change in practices.

Response: We are very pleased and encouraged to receive your positive comments regarding our research. We are really thankful and indebted by your critical comments and helpful suggestions. We believe that our responses would satisfy most of the questions rose by you and would highly appreciate your kind approval for its acceptance.

Abstract is clearly written but not of appropriate scientific tone, seems more written for translation of article to popular press. No specific data was provided in abstract.

Response: Thank you to mention this issue. We checked and modified this section following your suggestions by adding specific data to state clearly the scientific findings (page- 1; lines- 22 - 26).

Line 56:  I had to look up upazila, but seeing the definition I believe it is appropriate to continue using this term.  As it is used frequently, the meaning becomes clear.

Response: That is a very good point which we missed to define. Thanks again. There are five administrative tiers of local government in Bangladesh such as Division, District, Upazila, Union and Ward. The country is divided into 8 divisions, and each division is divided into districts which are further divided into sub-districts called ‘upazila’. Bangladesh has 495 upazilas. The upazilas are the third lowest tier of regional administration in Bangladesh. We added this information in the updated version (page- 2; lines- 62 - 64).

Line 87: dramatic language not needed in scientific paper.  Phrasing as “Considering the conditions of these vulnerable communities…” will be adequate.  The rest of the sentence also becomes confusing with the sensational language.  How can someone solve an unavoidable problem?  Perhaps just say that much research has been done on the problem and then get to literature review.

Response: Thanks a lot and we updated it according to your given suggestions (pages- 2 - 3; lines- 95 - 98).

Literature reporting is inadequate.  This paper states that study have been done but provides no relevant work to indicate how this study fits into the broader research.

Response: We do not disagree with you. We mentioned that most of the previous studies were about different issues (page- 3; lines- 98 - 102) rather than our objectives. However, we found only few studies that showed how the vulnerable communities can cope with waterlogging through agricultural multi-crop production (page- 3; line- 103 -104). These are the available literatures that we found to cite in this study.

Line 104: not clear why these document were collected or useful to the study.

Response: Before starting this study, we had to find out various documents and report the extent, frequency, and magnitude of waterlogging problem in those areas. This provided the background and logical reasons why we wanted to conduct this study. Only after that, we received the funds to conduct this study. Unfortunately, we did not keep our record of these literatures as they were from different local, national and international sources. To avoid confusion and follow the suggestion of reviewer-3, we removed this statement in the updated version (page- 3; lines- 117 - 119).

Line 109: What are unions?  Clarify relation of this term to an upazila?

Response: We clarified it now (page- 3; lines – 122 - 123). As mentioned above, the unions are the forth lowest tier of regional administration in Bangladesh.

Section 2.2 and 2.3 seem conflicting to me.  2.2 states that one farmer with no pond was chosen while section 2.3 states that three farmers (some with ponds) were chosen.  This needs to be clarified, perhaps it was two aspects of the study.

Response: We selected one farmer’s house that had a homestead pond, while the selected other two houses had no pond of which one was selected as a research farm (Figure 2). Hope it is clear now (page- 3; lines- 133 – 137 and page- 4; lines- 150 - 153).

Method section does not state over what time period data was collected or over what timeframe expected differences would be seen.

Response: The time frame has been mentioned now in the updated version (page- 4; line- 150 and 157).

I am glad that production and food security increased with intervention, but the cost as compared to benefit is unclear.  Who will provide the cost of developing water retention ponds, vertical infrastructure for vegetables, etc.?  I get the sense that individual landowners could not afford such.  Net income is finally addressed in section 3.7.  However, it should be explored more, perhaps considering the time to benefit of investment in research practices.

Response: The research farmers were fully guided and economically supported in converting the homestead by digging out the soil for a pond and raising the pond dikes along with other parts of the homestead. The research farmers also received advice on adaptive and alternative agricultural production techniques. Once their homesteads were reconstructed according to the design, the we provided the adaptive and alternative agricultural production technologies (page- 4; lines- 179 -191). During the calculation of benefit, all the costs for production were also considered. Hope it is clear now.

Section 3 – why are results in terms of income instead of yield?  Is it common for these farmers to sell all products and purchase own food separately.  I feel like the amount of food kept on hand has been overlooked and therefore family size may be a confounding factor.  Also, how does income compare to expense?  Perhaps there is not a difference in net profit and therefore food stability would be a better metric than income.

Response: The researchers provided a prescribed format (Supplementary File S1; page- 5 and line- 224) to the NGO’s field officer who regularly monitored and collected the total production data of each crop (including the food kept for homestead use). Therefore, we did not consider family size as a confounding factor.

Since each farmer cultivated different varieties of crops, it was not possible to compare the yield which has different values. Therefore, we considered only the selling price (monetary evaluation) to compare their production performances. Other factors like farmer’s income as labor, expenses, etc. were included into the data if they invested these for this crop production.  

Line 364: What data supports the statements made on women’ status?  I think this would be better included as a discussion point on potential benefits of the reported economic data than reporting as data.

Response: Thanks to notify this. Yes, we also agree with you as we don’t have any data to support this statement. So, we removed this statement from there. Since we already discussed this issue at the end of our discussion (page- 8; lines- 384 - 386), we avoided the redundancy by adding the similar points.  

Reviewer 2 Report

Abstract:

Line 18: ……‘with the other part, excavated’

Line 40-44:… Moreover…… climate change.    The sentence is too long.

Discussion

Line 383-384: Figures 5 a and b, the research farmer experienced a reduction in their net income. Please explain.

Author Response

# Response to reviewer-2

Abstract:

Line 18: ……‘with the other part, excavated’

Response: Thanks for this correction. Done (page- 1; line- 18)

Line 40-44:… Moreover…… climate change.    The sentence is too long.

Response: OK, it has now been divided into two short sentences (pages- 1 - 2; lines- 45 – 49+).

Discussion

Line 383-384: Figures 5 a and b, the research farmer experienced a reduction in their net income. Please explain.

Response: Thanks, we have included some statements to explain this issue in the discussion (page- 21; lines- 518 - 521).

Reviewer 3 Report

Dear Colleagues:

The article by Moshiur Rahman, Tapan Kumar Chakraborty, Abdullah Al Mamun and Victor Kiaya and whose title is "Land and water based adaptive farming practices in variedly elevated homesteads to cope with waterlogging" discusses a topic of interest: the development of strategies to cope with waterlogging in one of the regions most affected by climate change. I believe the article is of interest for publication in Sustainability. However, the authors should make a few changes before acceptance:

Abstract

Authors should state the objective of the study in the abstract.

Introduction

The introduction section is well structured. However, in the first paragraph, the authors must indicate that the Earth's temperature is expected to rise 1.5°C and reference must be made to Agenda 2030 and the Sustainable Development Goals, as they are related to the study.

I suggest this reference:

Keyber, L., & Lenzen, M. (2021). 1.5 °C degrowth scenarios suggest the need for new mitigation pathways. Nature Communications, 12, 2676. https://doi.org/10.1038/s41467-021-22884-9.

Line 96-100: authors must be more precise with the objective of the research.

Materials and methods

Authors must place tables and figures that are cited in the materials and methods section near the place where they are first cited.

On the other hand, I did not see anywhere in the materials and methods section the level of confidence shown by the surveyed population, both in the pilot phase and in the extended phase.

Also, I have not seen anywhere in the materials and methods section a justification of the questionnaire design, nor if the questionnaire has been validated by a Committee of Experts. Nor have the authors indicated whether the questionnaire was open-ended or closed-ended. Likewise, I believe that the authors should include the questionnaire used as supplementary material so that readers can better understand the results of the research.

In section 2.1 you indicate that you reviewed a large number of documents on the subject: how many exactly?

In section 2.5, you indicate that you conducted post hoc tests. There are many such tests: What test did you perform exactly?

Results

Authors must indicate the figures after citing them in the text for the first time.

The main comments in the results section are for Figures 4-13. In the text they indicate the existence of statistically significant differences based on the p-value and the post hoc test performed. However, in these figures the authors do not indicate the significance or the ranges formed in each year/event. The authors also do not provide a statistical note indicating what type of analysis they have performed on each figure. The authors must indicate such data so that the reader can understand the statistical analysis and observe precisely the behavior in each event/year.

Discussion

In the discussion section, the limitations of the research have not been indicated. At the end of the discussion and before the conclusions and recommendations section, the authors must indicate the limitations of the investigation: number of samples, locality, etc.

On the other hand, the authors must relate in the discussion section the contribution of their results to the fulfillment of the Sustainable Development Goals of the 2030 Agenda.

Acknowledgement and conflict of interest

Authors must indicate their acknowledgements in the section indicated for this purpose. In addition, authors must indicate the existence or not of any conflict of interest.

Kind regards

Author Response

# Response to reviewer-3

Dear Colleagues:

The article by Moshiur Rahman, Tapan Kumar Chakraborty, Abdullah Al Mamun and Victor Kiaya and whose title is "Land and water based adaptive farming practices in variedly elevated homesteads to cope with waterlogging" discusses a topic of interest: the development of strategies to cope with waterlogging in one of the regions most affected by climate change. I believe the article is of interest for publication in Sustainability. However, the authors should make a few changes before acceptance:

Response: We are glad and encouraged to receive your positive comments regarding our research. We are really thankful and indebted by your critical comments and helpful suggestions. We believe that our responses below would satisfy most of the questions rose by you.

Abstract

Authors should state the objective of the study in the abstract.

Response: Sorry, we mentioned it. Perhaps you missed it. We mentioned the objective as “this study was conducted in the eight most-affected areas in order to enhance agricultural production by applying Land and Water-based adaptive and alternative Farming Practices (LWFP)” (page- 1; lines- 14 - 16).

Introduction

The introduction section is well structured. However, in the first paragraph, the authors must indicate that the Earth's temperature is expected to rise 1.5°C and reference must be made to Agenda 2030 and the Sustainable Development Goals, as they are related to the study.

I suggest this reference:

Keyber, L., & Lenzen, M. (2021). 1.5 °C degrowth scenarios suggest the need for new mitigation pathways. Nature Communications, 12, 2676. https://doi.org/10.1038/s41467-021-22884-9.

Response: Thank you so much for providing this important reference. We added this issue in the updated version (page- 1; lines- 38 - 39) and (page- 3; lines- 105 - 108).

Line 96-100: authors must be more precise with the objective of the research.

Response: Done and please find it in the updated version (page- 3; lines- 109 - 113).

Materials and methods

Authors must place tables and figures that are cited in the materials and methods section near the place where they are first cited.

Response: Unfortunately, we can’t do it as the journal’s format does not allow us to place these tables and figures near the place where they are first cited.

On the other hand, I did not see anywhere in the materials and methods section the level of confidence shown by the surveyed population, both in the pilot phase and in the extended phase.

Response: Sorry, we did not estimate it as we did not design our study to include this parameter.

Also, I have not seen anywhere in the materials and methods section a justification of the questionnaire design, nor if the questionnaire has been validated by a Committee of Experts. Nor have the authors indicated whether the questionnaire was open-ended or closed-ended. Likewise, I believe that the authors should include the questionnaire used as supplementary material so that readers can better understand the results of the research.

Response: Thanks. Please find the questionnaire in the supplementary file and also find the statement in page- 5 and lines- 240 - 242.

In section 2.1 you indicate that you reviewed a large number of documents on the subject: how many exactly?

Response: Unfortunately, we did not keep our record of these literatures as they were from different local, national and international sources. To avoid confusion and follow the suggestion of reviewer-1, we removed this statement in the updated version (page- 3; lines- 117 - 119).

In section 2.5, you indicate that you conducted post hoc tests. There are many such tests: What test did you perform exactly?

Response: A good question. We performed the ‘Tukey’ test of adjustment for the pair-wise comparisons. We added this information to clarify the readers (page- 6; lines- 258 - 259).

Results

Authors must indicate the figures after citing them in the text for the first time.

Response: As mentioned in the Methods section, we are unable to do it as the journal’s format does not allow us to place these tables and figures near the place where they are first cited.

The main comments in the results section are for Figures 4-13. In the text they indicate the existence of statistically significant differences based on the p-value and the post hoc test performed. However, in these figures the authors do not indicate the significance or the ranges formed in each year/event. The authors also do not provide a statistical note indicating what type of analysis they have performed on each figure. The authors must indicate such data so that the reader can understand the statistical analysis and observe precisely the behavior in each event/year.

Response: Since adding this significance level made the figure clumsy, we avoided them. However, we also agreed with you and therefore, we added it in the updated version (Figures 4 – 13, except Fig. 12). We provided the statistical notes and other information in the caption of each figure (pages- 11 - 20; lines- 400 - 471)

Discussion

In the discussion section, the limitations of the research have not been indicated. At the end of the discussion and before the conclusions and recommendations section, the authors must indicate the limitations of the investigation: number of samples, locality, etc.

Response: We already indicated the limitations of our study in section 5 ‘Conclusion and Recommendation’ (page- 23 and lines- 616 – 620). We think this is the right place to discuss it. To avoid repetition, we did not add it again as you suggested.  

On the other hand, the authors must relate in the discussion section the contribution of their results to the fulfillment of the Sustainable Development Goals of the 2030 Agenda.

Response: Since none of our objectives was to contribute for the fulfillment of SDGs of the 2030 Agenda, we did not discuss it in our study.

Acknowledgement and conflict of interest

Authors must indicate their acknowledgements in the section indicated for this purpose. In addition, authors must indicate the existence or not of any conflict of interest.

Response: Perhaps, the format sent to you by the journal did not contain these sections which we already put at the end and just before the Reference section (page- 24; lines- 667 - 675.

Round 2

Reviewer 1 Report

Thank you making improvements

Reviewer 3 Report

Dear authors,

I consider that after making the corrections, the article is apt for publication in the journal.

Best regards